# Clinical Value of CT-Guided Fine Needle Aspiration and Tissue-Core Biopsy of Thoracic Masses in the Dog and Cat

**DOI:** 10.3390/ani11030883

**Published:** 2021-03-19

**Authors:** Massimo Vignoli, Roberto Tamburro, Andrea Felici, Francesca Del Signore, Annalisa Dettori, Morena Di Tommaso, Angela Ghiraldelli, Rossella Terragni, Francesco Simeoni, Ilaria Falerno, Arianna Miglio

**Affiliations:** 1Faculty of Veterinary Medicine, University of Teramo, 64100 Teramo, Italy; rtamburro@unite.it (R.T.); fdelsignore@unite.it (F.D.S.); mditommaso@unite.it (M.D.T.); fsimeoni@unite.it (F.S.); ifalerno@unite.it (I.F.); 2Istituto Zooprofilattico Sperimentale dell’Umbria e delle Marche, 06126 Perugia, Italy; a.felici@izsum.it (A.F.); a.dettori@izsum.it (A.D.); 3Pet Care Veterinary Clinic, 40133 Bologna, Italy; ghiraldelli.angela@gmail.com (A.G.); terragni.rossella@gmail.com (R.T.)

**Keywords:** Computer Tomography, CT, fine-needle aspiration, cytology, tissue-core biopsy, histology, thoracic lesions, dog, cat

## Abstract

**Simple Summary:**

Diagnostic imaging is of paramount importance in the diagnosis of thoracic lesions. Radiology has traditionally been considered the diagnostic procedure of choice for these diseases in addition to a correct cytological and histopathologic diagnosis. In human medicine, Computed Tomography (CT) and CT-guided biopsy are used in the presence of lesions which are not adequately diagnosed with other procedures. In the present study, thoracic lesions from 52 dogs and 10 cats of different sex, breed and size underwent both CT-guided fine-needle aspiration (FNAB) and tissue-core biopsy (TCB). In this study, 59 of 62 histopathological samples were diagnostic (95.2%). Cytology was diagnostic in 43 of 62 samples (69.4%). General accuracy for FNAB and TCB were 67.7% and 95.2%, respectively. Combining the two techniques, the overall mean accuracy for diagnosis was 98.4%. CT-guided FNAB cytology can be considered a useful and reliable technique, especially for small lesions or lesions located close to vital organs and therefore dangerous to biopsy in any other way.

**Abstract:**

Diagnosis of thoracic lesions on the basis of history and physical examination is often challenging. Diagnostic imaging is therefore of paramount importance in this field. Radiology has traditionally been considered the diagnostic procedure of choice for these diseases. Nevertheless, it is often not possible to differentiate inflammatory/infectious lesions from neoplastic diseases. A correct cytological and histopathologic diagnosis is therefore needed for an accurate diagnosis and subsequent prognostic and therapeutic approach. In human medicine, Computed Tomography (CT) and CT-guided biopsy are used in the presence of lesions which are not adequately diagnosed with other procedures. In the present study, thoracic lesions from 52 dogs and 10 cats of different sex, breed and size underwent both CT-guided fine-needle aspiration (FNAB) and tissue-core biopsy (TCB). Clinical examination, hematobiochemical analysis and chest radiography were performed on all animals. In this study, 59 of 62 histopathological samples were diagnostic (95.2%). Cytology was diagnostic in 43 of 62 samples (69.4%). General sensitivity, accuracy and PPV for FNAB and TCB were 67.7%, 67.7% and 100% and 96.7%, 95.2% and 98.3%, respectively. Combining the two techniques, the overall mean accuracy for diagnosis was 98.4%. Nineteen of 62 cases showed complications (30.6%). Mild pneumothorax was seen in 16 cases, whereas mild hemorrhage occurred in three cases. No major complications were encountered. CT-guided FNAB cytology can be considered a useful and reliable technique, especially for small lesions or lesions located close to vital organs and therefore dangerous to biopsy in other way.

## 1. Introduction

Diagnosis of thoracic lesions on the basis of history and physical examination is often challenging. Diagnostic imaging is therefore of paramount importance in this field. Radiology has traditionally been considered the elective diagnostic procedure for these diseases. Nevertheless, it is often not possible to differentiate inflammatory/infectious lesions from neoplastic disease. A correct cytological and histopathologic diagnosis is therefore needed for an accurate diagnosis and subsequent prognostic and therapeutic approach [1,2,3]. Other imaging modalities such as fluoroscopy, Ultrasonography (US), Computerized Tomography (CT) and Magnetic Resonance (MRI) have to be considered for the possibilities that they offer to the interventional radiologist to take guided biopsy samples [1,4,5,6]. US-guided fine needle aspiration biopsy (FNAB) or tissue core biopsy (TCB) of intrathoracic masses adjacent to the thoracic wall have been described in human medicine [7,8,9,10] as well as veterinary medicine [1,3]. Furthermore, the use of the Doppler examination allows the evaluation of the lesion vascularization [1]. Nevertheless, in human medicine, CT and CT-guided biopsies, both FNAB and TCB, are recommended in the presence of thoracic lesions which are not adequately visible or accessible with other procedures [11,12,13,14,15,16,17,18,19,20,21,22]. CT-guide sampling has been reported to be diagnostic in 75–95% of human patients [10,19,20,21]. On the other hand, indications for CT-guided biopsy in dogs and cats include lesions not well identified on radiographs or with fluoroscopy, lesions close to vascular structures, small lesions and lesions surrounded by air or gas which would not be identified with US [23,24,25,26]. In veterinary medicine, a few studies have been published on the CT-guided biopsy of the brain with stereotactic devices, while the description of the free hand technique CT-guided biopsy in small animals is still limited and not deeply investigated [23,24,26]. Particularly, few details and results are available about the sensitivity of the technique in different type of lesions. In one study, the accuracy of the CT-guided biopsy in the skeleton associated diseases was described [24]; TCB had an accuracy of 100% both for inflammatory/infectious and neoplastic diseases, while FNAB had an accuracy of 83.3%, with an overall mean accuracy of 95.7%. In one report, on intrathoracic masses, the CT-guided biopsy had an accuracy of 83% for TCB and 65% for FNAB [23]. However, no investigation has been reported on the usefulness of CT-guided FNAB compared to TCB. The aim of this paper is to describe and compare the diagnostic results of CT-guided thoracic FNAB and TCB in a large sample of animals and assess the percentage of diagnostic samples as well as the incidence and severity of associated complications. To the authors’ knowledge, this is the first study comparing FNAB and TCB under CT guidance.

## 2. Materials and Methods

### 2.1. Animals

In this retrospective study, 62 animals, 52 dogs and 10 cats of different breed, sex and size, underwent free hand technique CT-guided biopsy of the thoracic masses at Pet Care Veterinary Clinic (Spilamberto (MO), Bologna, Italy), from 2014 to 2020. All the CT reports were reviewed. Inclusion criteria were the selection of animals that underwent a CT examination, followed by both CT-guided FNAB and TCB, and had reached a final histopathological diagnosis of the lesions, serving as the gold standard, obtained by post-surgical or necropsy histopathological samples or by the TCB results subsequently confirmed by clinical and therapeutic follow up. Only 2 specimens (1 from FNAB and 1 from TCB) were taken in each lesion, in order to limit possible complications. Before the procedure, all animals were checked with blood and urine examinations. Thoracic radiograph was always performed before CT examination. The biopsies were taken by a board-certified radiologist (M.V.).

All animals were studied under general anesthesia and monitored during the procedure. For all examinations, there was the approval of the owners by informed consent signature. All clinical procedures and the care of the animals complied with the national legislation on animal care (Legislation decree n.26, 03/03/2014) and adhered to the internal rules of University of Teramo.

### 2.2. CT Examinations and Biopsy Collection

The CT examinations were performed to assess the extent of the lesion, diagnose eventual metastases and take an aimed biopsy. For the CT studies, a multidetector CT was used (BrightSpeed GE and Optima 540 GE, Milwaukee; WI, USA). The gantry was never tilted and the slice thickness was 1.25–2.5 mm. The CT was repeated after i.v. contrast medium (Optiray, Guerbet, Roissy, France) administration at the dose of 600 mg/kg i.v. The CT study was reviewed with lung (WW 1500, WL—550), soft tissue (WW 300–350, WL 35–40) windows and bone window (WW 1500 WL 450) when a rib was involved. Then with the same soft tissue window the biopsy was performed. In all cases FNAB was taken first and then TCB, being FNAB less traumatic and therefore less likely to cause a complication, e.g., pneumothorax. In some cases, to decrease the streak artifacts, a bone window (WW 2000 WL 450) was obtained. For FNAB, a 90 mm long, 21 gauge (G) spinal needle (Ago spinale, Artsana. Cuneo, Italy) was used. For the TCB, a 14 or 16 G spring loaded automated needle, with 15–23 mm of excursion (Angelo Franceschini, S. Lazzaro, BO, Italy), or 100 mm long, 12 G calibrated bone biopsy needle (Ago biopsia, Gallini S.R.L., Mirandola, MO, Italy), was used. The automated and bone needles were calibrated at 1 cm. All animals were positioned in sternal recumbency for the whole-body scan within a foam cradle, and the same position was used for the biopsies. For soft tissue diseases, when the lesions were located in deep ventral part of the thorax, seen on the base of thoracic radiographs, the foam cradle was not used, and the dog was positioned directly on the CT table to allow the biopsy with the same position. After the CT study was completed, the assessment of the location and extent of the lesion and the selection of the target plane was done. The target plane was chosen in an area with significant changes to obtain viable tissue samples. Areas suspected to be necrotic (with no contrast enhancement) and large vessels were avoided. Then, the CT table was moved to the target plane which was indicated by the laser light in the gantry (Figure 1). In this plane, the site for insertion of the needle was subjectively chosen and marked with a sterile radiopaque metal marker or syringe needle inserted a few millimeters into the chest wall, after surgical preparation was done. Subsequently, a few additional slices in the area of the marker were acquired to measure the distance from the skin to the superficial and deep borders of the lesion and to the area to biopsy. Those measurements facilitated the choice of correct depth and the angle of insertion of the needle. The CT table was moved out of the gantry so that the needle could be placed and advanced in the preset distance and angle after a skin incision. The position of the needle was evaluated with additional images and correction of needle placement was performed when indicated before the lesion was sampled. For FNAB, once the needle was in a correct position, the stylet was retracted and suction with a syringe was applied. For TCB, when the automated needle was correctly inserted, a tissue-core biopsy was obtained. Some more images in the area of the lesion were taken in order to check for complications. In the case of bone biopsy for rib lesion, the technique is similar [24], All animals were clinically monitored after the procedure for 2 h depending on eventual presence of complications (pneumothorax and pulmonary hemorrhage). As inclusion criteria, the animals selected must have underwent post-surgical or necropsy histopathological examination to obtain a final pathological diagnosis of the lesions, serving as the gold standard. Recorded data included signalment (breed, age and sex), localization of lesion, FNAB and TCB diagnosis, final histological diagnosis and any possible complication. Particularly, the degree of pneumothorax and pulmonary hemorrhage was subjectively scored by an author (M.V.) as mild, moderate or severe. Cytological and histological diagnosis were listed as not diagnostic when the interpretation was inconclusive (Table 1).

### 2.3. Statistical Analysis

Accuracy of cytology and TCB-obtained histological results was assessed by concordance with final histological diagnosis results. To present the true/false positive and negative results for carcinoma, inflammatory/infectious, lymphoma, mesothelioma, sarcoma and thymoma conditions diagnosed by two methods, 2 × 2 tables were used. Table 2 shows calculations of sensitivity, accuracy and positive predictive value (PPV or precision) for both diagnostic tools regarding diseases. Moreover, weighted arithmetic (revealing the accuracy values that take into account different contributions of the diseases in the study) mean sensitivity, accuracy and PPV, respectively, for FNAB and TCB were calculated. General sensitivity, accuracy and PPV were also calculated evaluating for each condition the concordance/discordance with the final diagnosis (for FNAC and TCB separately).

## 3. Results

The results of signalment, location of the lesion, FNAB/TAB diagnosis, final diagnosis and complications that occurred for each animal are reported in Table 1. The tip of the needle was visualized within the lesion in all patients (Figure 2 and Figure 3). Sixty-two animals had a CT-guided FNAB and TCB biopsy of intrathoracic lesion, 52 from dogs and 10 from cats. Thirty-five of 62 lesions were located in the lung, 18 were situated in the mediastinum, 7 in the thoracic wall and 1 the pleura. The pulmonary lesion distribution included 18 left and 17 right lungs lesions, and 23 of these were distributed in the caudal lobes, 11 in the cranial lobes, 4 in the middle lung lobes and 1 in the accessory lobe. At final histopathological diagnosis, masses were classified as 29 carcinomas, 11 sarcomas, 11 lymphomas, 6 thymomas, 4 inflammatory/infectious and 1 mesothelioma. Figure 4 shows 2 × 2 tables showing true/false positive and negative results for all conditions diagnosed and Table 2 shows the sensitivity, specificity, accuracy and PPV (precision) for TCB and FNAB, as well as for both diagnostic tools regarding diseases. Forty-one of 62 cases (66.1%) had concordant FNAB and TCB results, 38 (61.3%) were neoplastic masses and 3 (4.8%) were inflammatory/infectious lesions. Two of 62 cases (3.2%) had discordant FNAB and TCB results, one was diagnostic in histopathology not with cytology (mesothelioma) and one was diagnostic with cytology and not in histopathology. The latter was confirmed with necropsy and histopathology after death, being a sarcoma as diagnosed with FNAB. In one of 52 dogs (1.6%), TCB was not diagnostic because of only the presence of fibrous tissue. In the same dog, the FNAB was diagnostic for carcinoma and the diagnosis was confirmed after surgery. In one of 52 dogs (1.6%), both FNAB and TCB resulted non-diagnostic because only blood was present in cytology and fibrous tissue in histopathology. After necropsy and subsequent histopathology, a lung carcinoma was diagnosed. Seventeen of 62 cases (27.4%) had diagnostic results only for TCB. The diagnosis was reached in 43 of 62 FNAB (69.4%) and 59 of 62 TCB (95.2%) (Table 1). Eighteen of 62 FNAB (30.6%) and three of 62 TCB (4.8%) were not diagnostic because only blood or insufficient tissue were sampled, and one of 62 FNAB (1.6%) had an incorrect cytological diagnosis, reactive mesothelium, instead of mesothelioma. All TCB specimens that were of appropriate diagnostic quality provided a correct diagnosis. General sensitivity, accuracy and PPV for FNAB were 67.7%, 67.7% and 100%. General sensitivity, accuracy and PPV for TCB were 96.7%, 95.2% and 98.3%. Weighted arithmetic means (sensitivity, accuracy and PPV) that take into account the different contributions of the diseases in the study were 68%, 68% and 100% for FNAB and 95%, 94% and 99% for TCB. Combining the two techniques, the overall mean accuracy for diagnosis was 98.4%.

Nineteen of 62 patients showed procedural complications (30.6%), identified only on CT-images. Mild to moderate pneumothorax was present in 16 of 62 cases (25.8%) of thoracic biopsies TCB, and only one case after FNAB. In two cases, a mild lung hemorrhage was present after TCB (4.8%). The deeper was the lesion, the more frequent was the pneumothorax. No clinical manifestations occurred and none of the complications needed a surgical intervention. 

Neoplastic lesions were detected in 57 of 61 cases that reached a CT-guided biopsy diagnosis (93.4%; 49 dogs and 8 cats); 28 were carcinomas, 11 were lymphoma, 9 were sarcomas, 6 were thymomas, 2 were histiocytic sarcomas and 1 was mesothelioma (Figure 3). Non-neoplastic lesions (inflammatory/infectious lesions and abscess) were detected in cases (6.6%, two dogs and t cats). Thoracic lesions were located in the lung (35/62, 56.4%; 27 carcinomas, 3 inflammatory/infectious diseases, 2 histiocytic sarcoma, 1 metastatic hemangiosarcoma, 1 metastatic sarcoma and 1 non diagnostic), mediastinum (18/62, 29%; 11 lymphomas, 6 thymomas and 1 thymic carcinoma), thoracic wall (8/62, 12.9%; 5 fibrosarcomas, 1 osteosarcoma and 1 chondrosarcoma of the ribs and 1 inflammatory/infectious) and pleura (1/62, 1.6%; 1 mesothelioma). Three cases (Nos. 13, 48 and 62) with lung granuloma/abscess were surgically treated and the final histopathology confirmed the benign origin of the lesion. One cat (No. 54) with lung inflammatory/infectious mass was treated with antibiotics and went to final healing, clinically followed.

## 4. Discussion

In this study, free hand percutaneous CT-guided FNAB and TCB were performed in thorax lesions with the aim to compare diagnoses obtained by TCB and FNAB results to final histopathological findings, serving as the gold standard.

The reason to perform this study is that CT of the thorax is a very sensitive but not very specific method for diagnosis of focal lesions, even with the use of contrast medium. Particularly, a study on five cats showed that the density of the lesion measured with Hounsfield units is not specific for a neoplastic disease versus an inflammatory/infectious disease and that contrast medium does not give different enhancement [2]. Therefore, a biopsy is needed to establish a final diagnosis. Biopsy for superficial lesion could also be taken with ultrasound (US) or fluoroscopic guide. However, CT allows better evaluation of the extent of the lesion than US and fluoroscopy [24], as well as in lesions surrounded by gas as opposed to in US [1,24]. Moreover, CT is a more sensitive technique to examine for metastases compared to fluoroscopy and conventional radiology [7,27]. The results of our study show that FNAB reached the final diagnosis in 69.4% of the cases versus 95.2% of TCB with a general sensitivity, accuracy and PPV of 67.7%, 67.7% and 100%, respectively, for FNAB and of 96.7%, 95.2% and 98.3%, respectively, for TCB. In addition, when both techniques were combined, the overall mean accuracy was 98.4%.

These results are higher than those previously found by Zekas et al. [23] who reported an accuracy of 65% for FNAB and 83% for TCB in 31 cases. Tidwell and Johnson (1994) [25] described five patient with intrathoracic diseases (four lung and one cranial mediastinal lesions) reporting that four biopsy samples evaluated for cytology and three for histopathology were diagnostic [25]. In another study [2] describing CT-guided intra-thoracic lesions in four cats, only FNAB were performed and they revealed carcinoma in three of four cases (75%). The limitations of these previous studies were the evaluation of few cases, the lack of performing both types of biopsies for each lesion and the absence of surgery or necropsy that allow the final histopathological diagnosis for uncertain cases. Conversely, we performed both techniques for each case and, having taken only one biopsy sample with each technique, we believe that the results of our study can be considered satisfactory.

In human medicine, the diagnostic accuracy of the core biopsy under CT guidance is reported between 88% [16] and 95% [13], with similar results for FNAB, 85% [15]. In the literature, it has been reported that carcinomas and round cell tumors exfoliate much better than sarcomas [1], and, since most of the malignancies reported in the lung are carcinomas [20] and in the mediastinum are lymphomas or thymomas [28,29,30,31], this may explain why there is quite a high accuracy with CT-guided FNAB in our study. Indeed, 29 cases were epithelial neoplasms, 28 from lungs and 1 from thymus. Most lesions in our study were singular pulmonary masses or one of several nodules/masses in an animal with known neoplasia at another site (spleen and liver), and the final diagnosis was reached in 28 cases; 18 of them were diagnostic with both techniques, 27 with TCB and 1 only with FNAB. In one case (No. 4), FNAB and TCB were both non diagnostic.

In our study, the 18 cases of mediastinal masses or enlarged mediastinal lymph nodes were tumors of the hemolymphatic system (11 lymphomas, 6 thymomas and 1 thymic carcinoma). Eleven of these (nine lymphoma and two thymomas) were diagnosed with both FNAB and TCB and the other seven (four thymomas, two lymphomas and one thymic carcinoma) with TCB alone. In the latter cases, CT-guided FNAB cytology was not diagnostic because the samples had blood contamination due to the nature of the neoplasia. Two other cases of hemolymphatic neoplasms (histiocytic sarcoma) were identified in the lung, one correctly diagnosed by TCB and FNAB and one only by using TCB, since cytology allowed only to categorize the lesion as a round cell tumor.

In our study of nine neoplastic lesions with a final diagnosis of mesenchymal neoplasia (seven thoracic wall and two lungs), both FNAB and TCB were diagnostic in seven cases, and, considering the combination of the two techniques, the diagnosis was reached in all cases. In almost all cases in which the lesions involved the thoracic wall (six of seven cases), both FNAB and TCB allowed diagnosing sarcoma, because the lytic ribs allowed easily taking the FNAB, except in one case. In one case of lung lesions (No. 45), cytology was instead predictive of sarcoma unlike TCB histopathology, which highlighted an inflammatory lesion, probably because an inadequate part of the lesion was biopsied. From our results, we can state that FNAB and TCB have been shown to be useful techniques in diagnosing thoracic mesenchymal tumors.

In the case of lesions of the pleura, where cytology has low ability to differentiate reactive mesothelium from mesothelioma, it is preferable to perform only TCB; in fact. in our case. the final diagnosis of mesothelioma was not reached by cytology due to the intrinsic low sensitivity of cytology to identify these tumors [32,33,34]. In the case of pleural diseases, it would be correct also consider the thoracoscopic biopsy as an appropriate alternative to image-guided biopsies [35].

In the four cases of inflammatory/infectious lesions (Nos. 13, 48, 54 and 62), three were identified in the lungs (one dog and two cats) by both FNAB and TCB TC-guided samples, whereas, in one case of inflammatory purulent lesion of the chest wall, only TCB was able to reach the diagnosis even if the lesion was superficial. FNAB and TCB have been shown to be useful techniques in diagnosing thoracic inflammatory/infectious lesions even if, due to the low number of cases identified, this study is not appropriate for determining the true accuracy of CT-guided sampling for nonneoplastic diseases.

Of the 19 cases resulting non-diagnostic for FNAB, 10 were carcinoma (10/28, 35%), 4 thymomas (4/6, 66%), 2 lymphomas (2/11, 18%), 1 fibrosarcoma (1/11.9%) and 1 inflammatory/infectious (1/11.9%). Surprisingly, in carcinomas and thymomas, the percentage of non-diagnostic samples at the cytology was relatively high even if they are strongly exfoliative lesions [19]. In our study, this could be explained because in some cases carcinomas and thymomas, being intrathoracic lesions, can be more difficult to be biopsied than masses of the thoracic wall.

Based on our results, the use of FNAB CT-guided technique in case of thoracic masses is highly recommended given the easy and rapid execution, the reduced risk of complications, the low cost compared to surgical biopsies, and fast diagnosis with relatively high accuracy. In these lesions, cytology is almost sufficient in the case of non-neoplastic lesions to reach the diagnosis, but it is advisable to have confirmation with TCB sampling for histology because neoplastic lesions are frequent. In addition, FNAB is also fundamental for an extemporaneous cytological examination that could allow a rapid diagnosis, even if not complete, and, eventually, the sampling could be easily repeated to assess the cellularity and adequacy of the biopsy. This would allow for example to avoid a TCB and to go to surgery with limited diagnostic damage, and then obtain the definitive histological diagnosis. The number of biopsies taken for each mass has not been standardized. Although tumor dissemination along the biopsy pathway is rarely described in veterinary medicine, one case of tumoral seeding after lung FNAB has been reported, and therefore the limitation of the biopsies number should be considered [36].

The localization of the needle tip in the percutaneous CT-guided biopsy has been considered the key point for the success of the procedure. It is of paramount importance to differentiate between the true tip of the needle from the impression of the false tip, which is visible when the CT scan comprises only the angled needle. It has been reported that the “low density” artifact visible immediately adjacent to the distal part of the tip of the needle may create a false positive impression; therefore, the correct position of the needle must be determined by evaluating the shape and the distinctness of the tip rather than the “low density” artifact [25,26]. The choice of biopsy needle and the position of the animal depends on the localization, dimension and distance from the skin surface to the lesion [1], and we think also on the experience of the radiologist. In the present study, the patients were in sternal recumbency; however, if needed after the whole body CT for tumor staging, the position can be changed to a lateral recumbency to facilitate the biopsy sampling. It is important to avoid the study in lateral recumbency from the beginning to avoid lung collapse and consequent metastases being missed [27]. Changing the position of the patient also affect the time of the study which become longer because must be restarted from the beginning. For the needle choice, a 21 G spinal needle for FNAB was considered. A fine needle has been recommended to avoid aspiration of blood that was, indeed, a limitation for 19 FNAB that were not diagnostic mainly due to blood contamination. For TCB, a 14 or 16 G calibrated automated needle was used. The length of the needle and the weight of the handle of the needle can be a limiting factor; thus, it is necessary that the needle is completely introduced into the chest wall to prevent changes in position during the movement of the table for CT control. Another limitation of TCB is that the biopsy needle we used has 15–23 mm of excursion, and we considered it not possible to take a TCB in lesions smaller than 3–4 cm, depending on the location. This limitation for TCB is an important issue that makes FNAB and cytology more important to reach the diagnosis in some patients. We observed some complications such as 2 mild lung hemorrhages and 17 cases of mild to moderate pneumothoraxes, but the rate was lower than what was previously been reported [23]. The lower incidence was found with CT-guided percutaneous FNAB, and this result agrees with the reports in human medicine [19,20]. Nevertheless, no complications required further therapeutic intervention. Most complication were seen after TCB, probably due to the larger size of the needle compared to that of the needle used for FNA and biopsy of deep lesions; however, with some deep lesions, neither hemorrhage nor pneumothorax was observed. Those complications were not correlated to the size of the lesion. It has been reported that, the deeper is the lesion, the more severe the pneumothorax is likely to be, but no clinical manifestations are reported [23]. In the literature, complications have been reported with CT-guided transthoracic sampling in both humans and animals. The most commonly reported are pneumothorax, with a wide (9–61%) range of incidence reported, and pulmonary hemorrhages noted in fewer than 20% of patients, and, in most of the studies, they are reported as subclinical [19,23]. In one case (No. 50), the moderate pneumothorax created was probably the cause of not being able to obtain a diagnostic sample on TCB. In the same case, the FNAB was diagnostic for carcinoma.

Since passing through more lung tissue is needed to get into a deeper lesion, we can speculate that this could be the reason for the higher number of pneumothorax complications. However, we still cannot explain why the biopsy in some deep lesions did not create any pneumothorax. 

In human medicine, pneumothorax is the most common complication of percutaneous CT-guided lung biopsy and ranges from 8% to 61% [37,38]. In one study on 289 patients, it was reported that application of a thoracic drainage was necessary in 14% of cases. In the same study, it was reported that deeper lesions, which require a wider trajectory angle, were risk factors for pneumothorax [37]. The transthoracic needle biopsy can be performed with high-diagnostic rate also in patient with stable pneumothorax caused by other procedures [39]. Other factors reported to increase the risk of pneumothorax in humans include large needle diameter, increased number of puncture attempts, decreased size of lesion, prolonged procedure time, coughing and performing both TCB and FNAB [19].

Mild pulmonary hemorrhage occurred in two cases after lung TCB with negligible clinical significances since the bleeding was always self-limiting. This rate of incidence is lower than what has previously been reported in animals [23]. Animals included in this study did not have pre-CT coagulation profiles, since it has been demonstrated that normal coagulation parameters do not preclude hemorrhage [19,23]; moreover, masses biopsied, either by their nature or by the organs from which they came, were generally not considered at risk of bleeding or at least significant, as would be the case for a liver biopsy. Respiratory movements did not cause problems during the procedures in our study. However, it is important not to move the animal during the procedure to avoid losing the target so that one has to restart the examination. A limitation during the biopsy procedure may be the presence of a rib along the path chosen for the needle. Therefore, it is always necessary to consider this factor when choosing the entry point of the biopsy needle into the chest wall. One disadvantages of CT-guided biopsy compared to US-guided biopsy is the non-real time control of the needle tip. This limitation could be bypassed with real time fluoro-CT, which however represents a radiation hazard for the personnel. The time for the whole procedure is different depending on the size and location of the lesion, experience of the radiologist and CT machine available. With a spiral CT, the whole procedure (scanning and biopsy) takes 5–30 min.

Our research has some limitations such as the limited number of non-neoplastic cases and some types of neoplasms (thymoma and mesothelioma). Since this is a retrospective study, data may be biased, and some data could not be reviewed as the record was not available. Moreover, we could not control the impact of the proficiency of all operators on the results.

## 5. Conclusions

CT-guided biopsy is a safe and accurate technique. CT is useful for examination of areas that are difficult to reach with other techniques, especially in lesions surrounded by gas. Moreover, within the same examination, it is possible to assess the primary and secondary lesions and concomitant diseases. Based on our results, CT-guided percutaneous FNAB and TCB for thoracic lesions, particularly pulmonary, mediastinal and thorax wall lesions, are highly recommended given the easy and rapid execution, the reduced risk of complications, the low cost and the fast diagnosis with relatively high accuracy. Moreover, more than one sample could be taken with very low risk of complications, likely increasing the diagnostic accuracy of the procedure. A general limitation would be when a lesion is smaller than 3–4 cm, which cannot be biopsied with the automated needle but only with a fine needle Therefore, the use of cytology alone can be a valuable diagnostic aid in the case of small lesions that do not allow CT-guided TCB sampling. On the other hand, core needle TCB has a slightly higher complication rate but a superior diagnostic accuracy.

## Figures and Tables

**Figure 1 animals-11-00883-f001:**
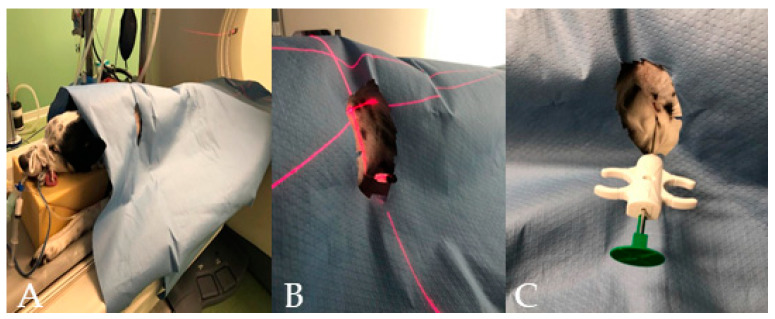
(**A**–**C**) Different phases of the CT–guided biopsy performed in a dog: (**A**) the dog on the CT table after the surgery was done; (**B**) the table moved to the selected slice and the laser light guiding the spinal needle insertion; and (**C**) the semi-automated tru-cut needle inserted into the thoracic wall.

**Figure 2 animals-11-00883-f002:**
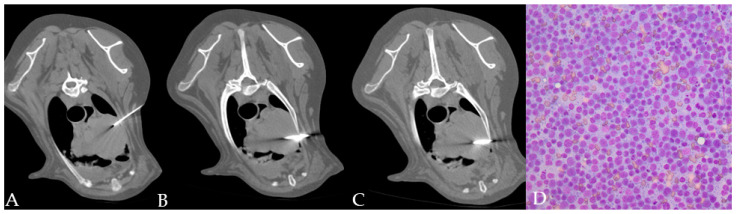
(**A**–**D**) CT-guided FNAB (**A**) and TCB (**B**,**C**) in a cranial mediastinal mass of a dog. The automated needle was inserted through the left thoracic wall obliquely in order to avoid the rib. Cytology allowed identifying a monomorphic population of lymphoid blasts ((**D**) 40x magnification) in a dog with a final diagnosis of lymphoma.

**Figure 3 animals-11-00883-f003:**
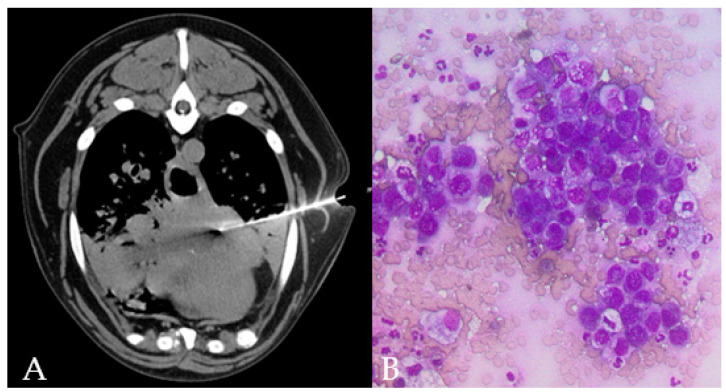
TC-guided FNAB in the accessory lung lobe of the right lung of a 15 year old, male Mix breed dog, is visible through the left thoracic wall (**A**). Lung collapse/infiltration of the ventral part of the cranial lung lobes is present. Cytology (**B**) shows clusters of epithelial cells with moderate pleomorphism and anisokaryosis, round nuclei eccentrically placed with multiple nucleoli and deeply basophilic cytoplasm with punctate vacuolations in a dog with final diagnosis of lung metastatic carcinoma.

**Figure 4 animals-11-00883-f004:**
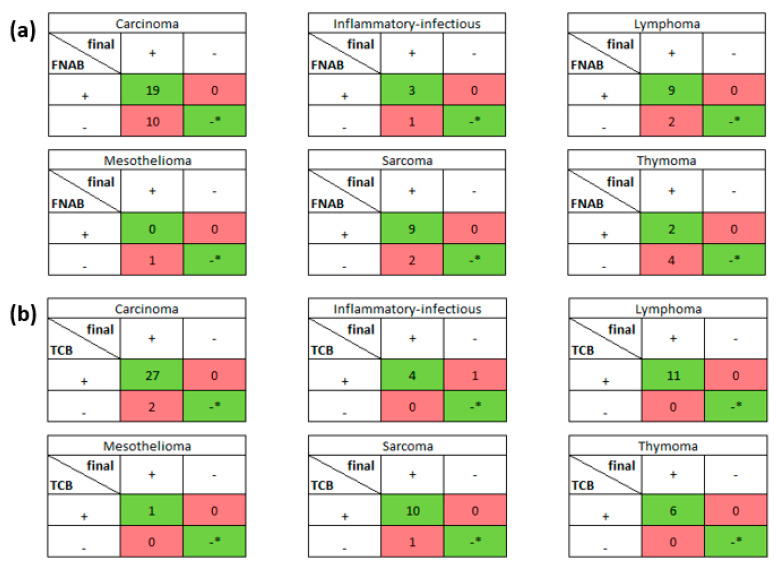
The tables show the numbers of obtained diagnoses for every disease separately for (**a**) FNAB and (**b**) TCB results. All 62 TCB and 62 FNAB results were compared with final histological diagnosis. The red boxes contain false negative and false positive results, whereas the green boxes contain true positive and negative (* no true negative was detected for all diseases) results.

**Table 1 animals-11-00883-t001:** CT-procedure and biopsy (FNAB/TCB) results of 52 dogs and 10 cats with thoracic lesions.

Cases Number	Signalment: Breed, Age in Years (y), Sex (M: Male; F: Female)	Location of the Lesion in the Thorax	Cytological (FNAB)/Histological Diagnosis (TCB)	Final Histological Diagnosis	Complications (Pneumothorax/Hemorrhages)
1	Maltese, 10 y, F	Thorax wall	Fibrosarcoma/Fibrosarcoma	Fibrosarcoma	
2	Italian Hound, 5 y, F	Pleura	Reactive mesothelium/Mesothelioma	Mesothelioma	
3	Labrador Retriever, 8 y, M	Lung (Left caudal lobe)	Carcinoma/Carcinoma	Carcinoma	Mild pneumothorax
4	Dogue de Bordeaux, 6 y, M	Lung (Left caudal lobe)	Not diagnostic/Not diagnostic	Carcinoma	Moderate pneumothorax
5	Jack Russell Terrier, 6 y, F	Lung (Right cranial lobe)	Carcinoma/Carcinoma (metastases from pancreas)	Carcinoma	Mild pneumothorax
6	Dobermann, 6 y, M	Lung (Right caudal lobe)	Sarcoma/Hemangiosarcoma (metastases from spleen)	Hemangiosarcoma	Moderate pneumothorax
7	Newfoundland, 12 y, F	Mediastinum (Lymph nodes)	Lymphoma/Lymphoma	Lymphoma	
8	Labrador Retriever, 9 y, M	Mediastinum	Lymphoma/Lymphoma	Lymphoma	
9	Dobermann, 9 y, F	Mediastinum	Lymphoma/Lymphoma	Lymphoma	
10	Mix breed, 6 y, M	Thorax wall (rib)	Sarcoma/chondroblastic osteosarcoma	Chondroblastic osteosarcoma	
11	Mix breed, 14 y, M	Lung (Left caudal lobe)	Carcinoma/Carcinoma	Carcinoma	
12	Dogue de Bordeaux, 8 y, M	Thorax wall	Not diagnostic/fibrosarcoma	Fibrosarcoma	
13	Coton de Tulear, 8 y, M	Thorax wall	Not diagnostic/Inflammatory (granuloma)	Inflammatory (granuloma)	
14	Mix breed, 12 y, M	Lung (Right middle lobe)	Not diagnostic/Carcinoma	Carcinoma	
15	Rottweiler, 8 y, M	Thorax wall (rib)	Mesenchymal neoplasia/chondrosarcoma	chondrosarcoma	
16	Pinscher, 12 y, F	Lung (Right caudal lobe)	Not diagnostic/Carcinoma	Carcinoma	
17	Mix breed, 12 y, F	Lung (Left caudal lobe)	Not diagnostic/Carcinoma	Carcinoma	
18	Labrador Retriever, 8 y, M	Mediastinum (Lymph-nodes)	Lymphoma/Lymphoma	Lymphoma	
19	Dwarf Poodle, 12 y, F	Lung (Left caudal lobe)	Carcinoma/Carcinoma	Carcinoma	
20	Cavalier King Charles Spaniel, 2 y, M	Mediastinum	Not diagnostic/Thymoma	Thymoma	
21	German Shepperd, 12 y, F	Mediastinum	Lymphoma/Lymphoma	Lymphoma	
22	Mix breed, 10 y, F	Thorax wall	Mesenchymal neoplasia/Fibrosarcoma	Fibrosarcoma	
23	Mix breed, 9 y, M	Thorax wall	Fibrosarcoma/Fibrosarcoma	Fibrosarcoma	
24	Boxer, 8 y, F	Lung, (Right cranial lobe)	Carcinoma/Carcinoma	Carcinoma	
25	Italian Hound, 12 y, M	Mediastinum	Thymoma/Thymoma	Thymoma	
26	Lhasa Apso, 11 y, M	Lung (Right cranial and caudal lobes)	Not diagnostic/Carcinoma	Carcinoma	
27	Cavalier King Charles Spaniel, 10 y, M	Mediastinum	Lymphoma/Lymphoma	Lymphoma	
28	Medium Schnauzer, 5 y, F	Mediastinum	Lymphoma/Lymphoma	Lymphoma	
29	Mix breed, 12 y, F	Lung (Right caudal lobe)	Carcinoma/Carcinoma	Carcinoma	
30	Dwarf Poodle, 8 y, M	Lung (left caudal lobe)	Not diagnostic/Carcinoma	Carcinoma	
31	Pitbull, 12 y, F	Lung (Right cranial lobe)	Carcinoma/Carcinoma	Carcinoma	
32	Mix breed, 11 y, M	Lung (Right middle lobe)	Not diagnostic/Carcinoma	Carcinoma	Mild pneumothorax
33	Mix breed, 11 y, F	Lung (Left caudal andcranial lobes)	Carcinoma/Carcinoma	Carcinoma	Mild pneumothorax
34	Bernese Mountain dog, 6 y, M	Lung (Left caudal lobe)	Round cell neoplasia/ Histiocytic Sarcoma	Histiocytic sarcoma	
35	Mix breed, 12 y, F	Lung (Left caudal lobe)	Not diagnostic/Carcinoma	Carcinoma	
36	Mix breed, 7 y, F	Lung (Left cranial lobe)	Carcinoma/Carcinoma	Carcinoma	Mild pneumothorax
37	Mix breed, 13 y, F	Mediastinum	Not diagnostic/Thymoma	Thymoma	
38	Italian Spinone dog, 11 y, F	Lung (Right middle lobe)	Histiocytic Sarcoma/ Histiocytic sarcoma	Histiocytic sarcoma	Moderate pneumothorax
39	Pitbull, 13 y, F	Mediastinum	Not diagnostic/Lymphoma	Lymphoma	
40	Mix breed, 15 y, M	Lung (Right accessory lobe)	Not diagnostic/Carcinoma	Carcinoma	Mild pneumothorax; mild hemorrhage
41	Mix breed, 10 y, M	Lung (Left cranial lobe)	Carcinoma/Carcinoma	Carcinoma	Mild pneumothorax
42	Mix breed, 13 y, F	Mediastinum	Not diagnostic/Lymphoma	Lymphoma	
43	Cirneco dell’Etna, 9 y, F	Mediastinum	Not diagnostic/Thymoma	Thymoma	
44	Mix breed, 14 y, F	Lung (left cranial lobe)	Carcinoma/Carcinoma	Carcinoma	Moderate pneumothorax
45	Mix breed, 10 y, M	Lung (right and middle cranial lobes)	Sarcoma/Inflammatory	Sarcoma	
46	Mix breed, 14 y, M	Lung (right caudal lobe)	Carcinoma/Carcinoma	Carcinoma	Mild pneumothorax
47	Mix breed, 4 y, F	Lung (right caudal lobe)	Carcinoma/Carcinoma	Carcinoma	Mild pneumothorax
48	Boxer, 9 y, M	Lung (left caudal lobe)	Abscess/Abscess	Abscess	
49	Mix breed, 12 y, F	Lung (left cranial lobe)	Carcinoma/Carcinoma	Carcinoma	
50	Bernese Mountain dog, 7 y, M	Lung (right caudal lobe)	Carcinoma/Not diagnostic	Carcinoma	Moderate pneumothorax
51	Dobermann, 12 y, M	Lung (left caudal lobe)	Carcinoma/Carcinoma	Carcinoma	Mild pneumothorax
52	Labrador Retriever, 9 y, F	Lung (left caudal lobe)	Carcinoma/Carcinoma	Carcinoma	
53	European Shorthair cat, 2 y, F	Mediastinum	Thymoma/Thymoma	Thymoma	
54	European Shorthair cat, 13 y, M	Lung (Left cranial lobe)	Inflammatory-infectious/Inflammatory-infectious	Inflammatory/-infectious	
55	European Shorthair cat, 10 y, F	Mediastinum	Lymphoma/Lymphoma	Lymphoma	
56	European longhair cat, 8 y, F	Thorax wall	Fibrosarcoma/Fibrosarcoma	Fibrosarcoma	
57	European Shorthair cat, 11 y, F	Lung (right caudal lobe)	Carcinoma/Carcinoma	Carcinoma	
58	European Shorthair cat, 9 y, M	Mediastinum	Not diagnostic/Thymic Carcinoma	Thymic Carcinoma	
59	European Shorthair cat, 5 y, F	Mediastinum	Not diagnostic/Thymoma	Thymoma	
60	European Shorthair cat, 12 y, F	Lung (Right caudal lobe)	Carcinoma/Carcinoma	Carcinoma	Mild pneumothorax
61	European Shorthair cat, 2 y, F	Mediastinum	Lymphoma/Lymphoma	Lymphoma	
62	European Shorthair cat, 9 y, M	Lung (left caudal lobe)	Abscess/Abscess	Abscess	Moderate pneumothorax, mild hemorrhage

**Table 2 animals-11-00883-t002:** The sensitivity, accuracy and PPV for FNAB and TCB regarding all diseases. Weighted arithmetic means reveal the accuracy values of the different contributions of the diseases in the study.

**FNAB**	**Carcinoma** **(n = 29)**	**Inflammatory–Infectious** **(n = 4)**	**Lymphoma** **(n = 11)**	**Mesothelioma** **(n = 1)**	**Sarcoma** **(n = 11)**	**Thymoma** **(n = 6)**	**Weighted Aritmetic Mean**
Sensitivity	0.66	0.75	0.82	0	0.82	0.33	0.68
Accuracy	0.66	0.75	0.82	0	0.82	0.33	0.68
PPV	1	1	1	-	1	1	1
**TCB**	**Carcinoma** **(n = 29)**	**Inflammatory–Infectious** **(n = 4)**	**Lymphoma** **(n = 11)**	**Mesothelioma** **(n = 1)**	**Sarcoma** **(n = 11)**	**Thymoma** **(n = 6)**	**Weighted Aritmetic Mean**
Sensitivity	0.93	1	1	1	0.91	1	0.95
Accuracy	0.93	0.80	1	1	0.91	1	0.94
PPV	1	0.80	1	1	1	1	0.99

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
