# Peer review of "Clinical Value of CT-Guided Fine Needle Aspiration and Tissue-Core Biopsy of Thoracic Masses in the Dog and Cat"

_animals, 2021, doi:10.3390/ani11030883_

Round 1
Reviewer 1 Report
The MS is well written, the materials and methods are described in a clear manner. The discussion is appropriate, including the limitations related to the limited non-neoplastic cases and the presence of some type of neoplasm.
A moderate English revision is requested.
Author Response
REV 1
Comments and Suggestions for Authors
The MS is well written, the materials and methods are described in a clear manner. The discussion is appropriate, including the limitations related to the limited non-neoplastic cases and the presence of some type of neoplasm.
A moderate English revision is requested.
Thank you very much for your comment and for the appreciation of our study. We agree to revise the english language, and for that we ask that it will be done by the Editorial Office of Animals.
Reviewer 2 Report
In addition to the group of dogs, the inclusion of a small number of cats in the study appears to be superfluous. It seems to me that the study would be better if it only included dogs, while a new cat study including a larger number of cats could be done in the future to further investigate that specific question. However, please note that if the authors still wants to include cats in the study, for me that is fine as well.
In the Introduction paragraph lines 60, 62 refer to references 4-6 and 9,10. However, curiously there are no references to numbers 7,8.
In Figures 2. and 3. (lines 236, 243 respectively) there is no L and R lettering to orient the images as being either left or right.
In line 261 there are too many spaces between the end of the last sentence and the beginning of the next. (“. General…”).
In line 289 There is the same problem of too many spaces between words describing. CT-guides.
In 308-310 the words “Lymphoma” and “Thymomas” are written in both capital and lower case, which is not consistent in the paper.
Similarly, a last of consistency in show between line 236 in which number is written as “n”, while in line 243 number is abreviated to “n°”. I think that using “n°” is clearer for a general audience.
In lines 236 and lines 243 with the appropriate references 24. And 25. “Veterinary Radiology & Ultrasound” needs to be reduced to “Vet Radiol Ultrasound”.
Aside for a few formatting quibbles that need to still be cleared up, I regard this paper as being a very interesting and very well done study, and I give it my unreserved recommendation.
Author Response
REV 2
Comments and Suggestions for Authors
In addition to the group of dogs, the inclusion of a small number of cats in the study appears to be superfluous. It seems to me that the study would be better if it only included dogs, while a new cat study including a larger number of cats could be done in the future to further investigate that specific question. However, please note that if the authors still wants to include cats in the study, for me that is fine as well.
Thank you for this appropriate comment. However, considered that there are no extensive reports on cats regarding this technique (to our knowledge), and since the biopsy method is the same as for dogs, we would prefer to leave the group of cats as well. It is certainly true that more extensive work in this species will be needed.
In the Introduction paragraph lines 60, 62 refer to references 4-6 and 9,10. However, curiously there are no references to numbers 7,8.
It has been reviewed and references have been added in the test
In Figures 2. and 3. (lines 236, 243 respectively) there is no L and R lettering to orient the images as being either left or right.
By international convention in diagnostic imaging images the right goes to the left of the reader and so these images were presented. To clarify, we have added the location (Left) of the biopsy needle in the captions.
In line 261 there are too many spaces between the end of the last sentence and the beginning of the next. (“. General…”).
Correction has been made.
In line 289 There is the same problem of too many spaces between words describing. CT-guides.
Correction has been made.
In 308-310 the words “Lymphoma” and “Thymomas” are written in both capital and lower case, which is not consistent in the paper.
All the capitals have been corrected for each disease through the text, and it should be now consistent. Thank you for this comment.
Similarly, a last of consistency in show between line 236 in which number is written as “n”, while in line 243 number is abreviated to “n°”. I think that using “n°” is clearer for a general audience.
Correction has been made, now in line 231.
In lines 236 and lines 243 with the appropriate references 24. And 25. “Veterinary Radiology & Ultrasound” needs to be reduced to “Vet Radiol Ultrasound”.
Correction has been made.
Aside for a few formatting quibbles that need to still be cleared up, I regard this paper as being a very interesting and very well done study, and I give it my unreserved recommendation.
Thank you very much for your comment.
Reviewer 3 Report
Thank you for the submission of your manuscript evaluating the use of CT to guide FNAs and biopsies of thoracic masses. The information in this manuscript is useful and provides good justification for using fine-needle aspiration of these lesions.
Specific comments follow:
Lines 15 & 29: I don’t understand what you mean by radiology being an “elective” procedure for diagnosis of thoracic lesions.
Lines 21 & 38: The word “samples” should be added after “histopathological”.
Line 84: This isn’t the first veterinary study to address a comparison, as several cases in your reference #22 (Zekas et al) had both FNA and biopsy performed. Consider rewriting this sentence to specify that your study is the first to make comparisons to the degree that you did.
Lines 93-95: Is it standard procedure at your practice to sample thoracic masses with both FNA and biopsy if achievable?
Line 123: Should the “and” between the biopsy needle descriptions actually be “or”? Presumably only one or the other was used.
Line 131: Consider adding a sentence that specifies that the next few sentences are for soft tissue lesions, as line 147 then specifies that it begins a section for sampling of bone lesions.
Line 138: The phrase “superficial and deep” is probably more accurate than “proximal and distal” for intrathoracic lesions.
Line 168: As there are two different versions of histopathology (the core biopsy vs. the gold standard), consider adding “TCB-obtained” before the phrase “histological results”.
Line 171: It is confusing to reference Figure 4 here, as we have not even gotten to Figures 2 and 3. It probably isn’t really necessary to reference it until the results, which will provide appropriate ordering of the figures.
Line 183: Somewhere in the early part of the Results section you should specifically comment on how many cases had a gold standard diagnosis and how many were based on clinical resolution. Was there only one case in the second category?
Line 189: The word “interested” is not the correct word for this situation.
Lines 189-191: Were there no lesions in the accessory lobe, or are these lesions included in the caudal lobes category? Since you refer to middle lung lobes (in the plural), are you including lesions of the caudal part of the left cranial lung lobe in this category?
Lines 197-200: The phrasing of this sentence is confusing. When you state that one method was not diagnostic, it suggests that the sample was poor or inconclusive, when really it was incorrect. Consider changing “diagnostic” to comments about being correct or incorrect. Also consider adding what the incorrect diagnosis was in each of the two cases (or that the histopath sample was nondiagnostic).
Lines 207-209: You state here that only 3 cases of TCB failed to reach a diagnosis (all of them non-diagnostic samples), but based on Figure 4b there were 3 false negatives and 1 false positive (4 total incorrect diagnoses). In Table 1, I find 3 incorrect answers from TCB, 2 non-diagnostic samples and one incorrect call on a sarcoma. Please explain the discrepancy.
Line 210: Consider reiterating that the correct diagnosis for the reactive mesothelium case was mesothelioma.
Line 210: It may be helpful to explicitly state that all TCB specimens that were of appropriate diagnostic quality provided a correct diagnosis, if this is true.
Line 219: Please provide more information regarding pneumothorax and the depth of lesions. Do you have actual data for this claim? Did you run any statistical tests?
Lines 231-234: How is the diagnosis of granuloma/abscess different from the diagnosis of pulmonary inflammatory/infectious mass? The way it is phrased it sounds as if the two are different categories. If the real difference in the two groups is the treatment, and not the pathology itself, consider deleting the word “The” from the beginning of the first sentence (line 231) and changing the word “The” on line 233 to “One”. Also, how did you confirm that the lesion in n54 resolved? Since you are including this case without a gold standard diagnosis, more details are necessary.
Lines 238-240 (and general): Do you ever consider using ultrasound to guide sampling in cases like that in Figure 2? At my institution we usually use ultrasound for sampling following CT if the lesion is accessible with US (due to the real-time nature of the sampling), so I’m curious about your thoughts. (We also have the luxury of an US machine dedicated to the CT/MR suite, so that helps.)
Line 259 (Fig 4b): Consider commenting in the text that the TCB false-positive case in the inflammatory/infectious group was actually a sarcoma?
Lines 261-264 (Table 2): The commas should be replaced with decimal points.
Line 266 (Discussion in general): Consider re-ordering your Discussion to start with the key take-home message from your manuscript. Many readers won’t read the entire Discussion, so you don’t want the key points hidden after literature review.
Lines 267-324: Please break this into several paragraphs.
Line 267: Please add that CT is very sensitive but not very specific “for diagnosis of focal lesions”. Sensitivity and specificity only relate to specific applications of a test and will be different for different disease states. (For example, CT is very sensitive and specific for identification of pneumothorax, but less so for the cause of pneumothorax.)
Line 271: Reference #26 relates to pneumothorax. Is this the citation that you meant to put here?
Lines 280-284: You’re just repeating the results here. Consider highlighting the most important results that you want to focus on rather than a list of lots of numbers.
Line 326: Other research has showed that FNAs of mesothelioma may not yield an answer with a desirable frequency. You should cite some of this research, as here it sounds as if your results led to this conclusion, which certainly isn’t appropriate on the basis of a single case.
Author Response
REV 3
Comments and Suggestions for Authors
Thank you for the submission of your manuscript evaluating the use of CT to guide FNAs and biopsies of thoracic masses. The information in this manuscript is useful and provides good justification for using fine-needle aspiration of these lesions.
Thank you very much for your comment.
Specific comments follow:
Lines 15 & 29: I don’t understand what you mean by radiology being an “elective” procedure for diagnosis of thoracic lesions.
Diagnostic procedure of choice, it has been corrected in the text.
Lines 21 & 38: The word “samples” should be added after “histopathological”.
Correction done.
Line 84: This isn’t the first veterinary study to address a comparison, as several cases in your reference #22 (Zekas et al) had both FNA and biopsy performed. Consider rewriting this sentence to specify that your study is the first to make comparisons to the degree that you did.
In the study by Zekas et al., VRU, 2005, there was not comparison between TCB and FNA biopsies under guidance. Infact, out of 30 cases, 12 had FNA, 10 TCB, and only 8 cases had both TCB and FNA, In our study all the cases had both TCB and FNA, we therefore believe that the nice study from Zekas et al. it was not a real comparison, instead in our it is.
Lines 93-95: Is it standard procedure at your practice to sample thoracic masses with both FNA and biopsy if achievable?
Thank you for this comment. In our practice it is common to take both semples when possibile. The reason is that cytology can be done immediately and we can often have a fast, despite non complete, result. In the last period we changed a bit technique, and we roll up the biotpic samples on the slides for extemporaneous cytology, but we do not yet have a precise idea of the accuracy of the method. It could be a future study.
Line 123: Should the “and” between the biopsy needle descriptions actually be “or”? Presumably only one or the other was used.
Correction has been done.
Line 131: Consider adding a sentence that specifies that the next few sentences are for soft tissue lesions, as line 147 then specifies that it begins a section for sampling of bone lesions.
A sentence has been added as requested.
Line 138: The phrase “superficial and deep” is probably more accurate than “proximal and distal” for intrathoracic lesions.
Correction has been done.
Line 168: As there are two different versions of histopathology (the core biopsy vs. the gold standard), consider adding “TCB-obtained” before the phrase “histological results”.
Correction has been done.
Line 171: It is confusing to reference Figure 4 here, as we have not even gotten to Figures 2 and 3. It probably isn’t really necessary to reference it until the results, which will provide appropriate ordering of the figures.
Thank you for this comment, this was an oversight. The ref. Fig. 4 has been deleted.
Line 183: Somewhere in the early part of the Results section you should specifically comment on how many cases had a gold standard diagnosis and how many were based on clinical resolution. Was there only one case in the second category?
All the informations are in the text, and we think the non diagnostic samples should follow the diagnostics. We highlighted in the text, in red, the cases that required the "gold standard" histology, post mortem or post surgery (3 cases) in the current lines 210-211 and the only case that was clinically followed line 235- 236.
Line 189: The word “interested” is not the correct word for this situation.
Thank you for this comment, correction has been done.
Lines 189-191: Were there no lesions in the accessory lobe, or are these lesions included in the caudal lobes category? Since you refer to middle lung lobes (in the plural), are you including lesions of the caudal part of the left cranial lung lobe in this category?
Thank you for this comment. Accessory lobe was involved in one case, and added in the text.
Lines 197-200: The phrasing of this sentence is confusing. When you state that one method was not diagnostic, it suggests that the sample was poor or inconclusive, when really it was incorrect. Consider changing “diagnostic” to comments about being correct or incorrect. Also consider adding what the incorrect diagnosis was in each of the two cases (or that the histopath sample was nondiagnostic).
The informations requested are in the text, se actual lines 200-209, the text is highlighted in green.
Lines 207-209: You state here that only 3 cases of TCB failed to reach a diagnosis (all of them non-diagnostic samples), but based on Figure 4b there were 3 false negatives and 1 false positive (4 total incorrect diagnoses). In Table 1, I find 3 incorrect answers from TCB, 2 non-diagnostic samples and one incorrect call on a sarcoma. Please explain the discrepancy.
The incorrect diagnoses for TCB are 3, two cases of carcinoma and one case of sarcoma. The case of sarcoma was diagnosed in TCB as inflammatory and for this reason it was put in false positives, but in substance it was counted twice in this table. However, since this information is not extremely important for our results, if it is considered confusing we can remove it from the table.
Line 210: Consider reiterating that the correct diagnosis for the reactive mesothelium case was mesothelioma.
Correction has been done.
Line 210: It may be helpful to explicitly state that all TCB specimens that were of appropriate diagnostic quality provided a correct diagnosis, if this is true.
The sentence has been added, thank you for this comment.
Line 219: Please provide more information regarding pneumothorax and the depth of lesions. Do you have actual data for this claim? Did you run any statistical tests?
Thank you for this comment. Unfortunately we do not have statistical analysis of this data, it was just a subjective abservation.
Lines 231-234: How is the diagnosis of granuloma/abscess different from the diagnosis of pulmonary inflammatory/infectious mass? The way it is phrased it sounds as if the two are different categories. If the real difference in the two groups is the treatment, and not the pathology itself, consider deleting the word “The” from the beginning of the first sentence (line 231) and changing the word “The” on line 233 to “One”.
Actually abscess presents as a capsulated mass containing fluid, while granuloma is more solid (may contain viable tissue and/or necrotic tissue as described in the veterinary literature, and inflammatory/infectious such as pneumonia are more infiltrative lesion of the lung, not always possible to differentiate from a tumor, such as histiocityc sarcoma as in Fig. 3.
The corrections have been anyway done as requested.
Also, how did you confirm that the lesion in n54 resolved? Since you are including this case without a gold standard diagnosis, more details are necessary.
The informations are in the text, highlighted in red, and it has been added that the cat has been followed clincally after the therapy until complete remission occurred.
Lines 238-240 (and general): Do you ever consider using ultrasound to guide sampling in cases like that in Figure 2? At my institution we usually use ultrasound for sampling following CT if the lesion is accessible with US (due to the real-time nature of the sampling), so I’m curious about your thoughts. (We also have the luxury of an US machine dedicated to the CT/MR suite, so that helps.)
Thank you very much for this comment. I am also lucky to have an US machine in the CT and MRI room, Personally I feel more confrotable to biopsy thoracic lesions under CT-guidance, even if not with real time control of the needle as in US (I also have the CT Fluoro in real time eventually). Instrad, I only use US for taking abdominal or retroperitoneal biopsies, despite, at leas tin human being, especially for retroperitoneal masses are often biopsed under CT guidance. I think often it is a forma mentis rather than a real advantage. For deep lesions I think that US is not suitable. Last, for bony lesion biopsies, such as in the vertebral body or ribs masses, CT I think it is far superior.
Line 259 (Fig 4b): Consider commenting in the text that the TCB false-positive case in the inflammatory/infectious group was actually a sarcoma?
The information is in the discussion, highlighted in red, actual lines 325-327.
Lines 261-264 (Table 2): The commas should be replaced with decimal points.
Correction has been made.
Line 266 (Discussion in general): Consider re-ordering your Discussion to start with the key take-home message from your manuscript. Many readers won’t read the entire Discussion, so you don’t want the key points hidden after literature review.
Correction has been made as requested.
Lines 267-324: Please break this into several paragraphs.
Correction has been made as requested.
Line 267: Please add that CT is very sensitive but not very specific “for diagnosis of focal lesions”. Sensitivity and specificity only relate to specific applications of a test and will be different for different disease states. (For example, CT is very sensitive and specific for identification of pneumothorax, but less so for the cause of pneumothorax.)
Correction has been made as requested.
Line 271: Reference #26 relates to pneumothorax. Is this the citation that you meant to put here?
We apologize for this mistake, correction has been done.
Lines 280-284: You’re just repeating the results here. Consider highlighting the most important results that you want to focus on rather than a list of lots of numbers.
Correction has been made as requested.
Line 326: Other research has showed that FNAs of mesothelioma may not yield an answer with a desirable frequency. You should cite some of this research, as here it sounds as if your results led to this conclusion, which certainly isn’t appropriate on the basis of a single case.
Citations have been added as requested.